# Factors Influencing O_3_ Concentration in Traffic and Urban Environments: A Case Study of Guangzhou City

**DOI:** 10.3390/ijerph191912961

**Published:** 2022-10-10

**Authors:** Tao Liu, Jia Sun, Baihua Liu, Miao Li, Yingbin Deng, Wenlong Jing, Ji Yang

**Affiliations:** 1College of Geographical Science, Harbin Normal University, Harbin 150025, China; 2Guangdong Open Laboratory of Geospatial Information Technology and Application, Lab of Guangdong for Utilization of Remote Sensing and Geographical Information System, Guangzhou Institute of Geography, Guangdong Academy of Sciences, Guangzhou 510070, China; 3Southern Marine Science and Engineering Guangdong Laboratory (Guangzhou), Guangzhou 511485, China

**Keywords:** ozone, nitrogen dioxide, traffic condition, impact factors

## Abstract

Ozone (O_3_) pollution is a serious issue in China, posing a significant threat to people’s health. Traffic emissions are the main pollutant source in urban areas. NO_X_ and volatile organic compounds (VOCs) from traffic emissions are the main precursors of O_3_. Thus, it is crucial to investigate the relationship between traffic conditions and O_3_ pollution. This study focused on the potential relationship between O_3_ concentration and traffic conditions at a roadside and urban background in Guangzhou, one of the largest cities in China. The results demonstrated that no significant difference in the O_3_ concentration was observed between roadside and urban background environments. However, the O_3_ concentration was 2 to 3 times higher on sunny days (above 90 μg/m^3^) than on cloudy days due to meteorological conditions. The results confirmed that limiting traffic emissions may increase O_3_ concentrations in Guangzhou. Therefore, the focus should be on industrial, energy, and transportation emission mitigation and the influence of meteorological conditions to minimize O_3_ pollution. The results in this study provide some theoretical basis for mitigation emission policies in China.

## 1. Introduction

Air pollution has become a crucial issue in China due to rapid economic development [1]. The Chinese government has exerted a significant effort to reduce air pollution in recent years. As a result, fine particulate matter (PM_2.5_) has significantly decreased due to strict emission mitigation policies [2]. Ozone (O_3_) has become the most prevalent pollutant in China. The O_3_ concentration has increased by 10.6% from 2015 to 2021 in 339 [3,4]. Excessive exposure to O_3_ can be extremely harmful to human health, causing substantial damage and irritation to the eyes, respiratory tract, and lungs [5,6,7].

Many studies have focused on O_3_ pollution in China, investigating the spatiotemporal variations [8,9,10,11,12], secondary formation mechanism [13,14,15], emission sources [16,17,18,19], and other factors. The Pearl River Delta (PRD) is one of the most developed regions in China and has experienced significant O_3_ pollution. The O_3_ concentration has increased in the PRD since 2015 [20]. The O_3_ pollution is the highest in autumn in the PRD due to high temperatures, strong solar radiation, and low relative humidity (RH) [21,22,23,24,25]. In addition, several studies confirmed the “weekend effect” [26,27] in China, i.e., the O_3_ concentration is higher on weekends than during working days in Beijing [28], Shanghai [29,30], and Guangzhou [31]. There are two reasons. First, the nitric oxide (NO) concentration is lower during the weekend due to fewer traffic emissions. Therefore, the inhibitory effect of NO on O_3_ is weaker, and more O_3_ is generated. Second, fewer aerosol particles are emitted during the weekend, resulting in less scattering and absorption of solar radiation. As a result, more O_3_ is formed due to the stronger solar radiation during weekends [32].

There are three major sources of near-ground O_3_ precursors: traffic emissions [33], industry emissions, and emissions by power plants [34]. Mitigating O_3_ pollution has become a crucial issue in the PRD region in recent years [35]. However, it is challenging to control O_3_ pollution due to the complex O_3_ generation mechanism [36]. After absorbing ultraviolet light, tropospheric O_2_ decomposes into two O atoms. The O atoms are combined with O_2_ to form O_3_ (Equations (1) and (2)). In urban areas, NO_2_ in traffic emissions is the main precursor of O_3_ (Equation (3)). O_3_ rapidly oxidize NO to form NO_2_, known as the titration effect (Equation (4)).
(1)$$O_{2}+UV\to \;O+O$$
(2)$$O+O_{2}+M\;\to \;O_{3}+M$$
(3)$$NO_{2}+hv\;\to \;NO+O$$
(4)$$O_{3}+NO\;\to \;NO_{2}+O_{2}$$

In these processes, a dynamic equilibrium exists during the formation and consumption of O_3_ by NO_X_. However, alkoxy radicals (RO) and hydroperoxyl radicals (HO_2_) generated by the reaction of volatile organic compounds (VOCs) and hydroxyl (OH) radicals in the atmosphere also react with NO (Equations (5)–(8)), destroying the dynamic balance between NO_X_ and O_3_ and increasing the O_3_ concentration.
(5)$$HO_{2}+NO\;\to \;HO+NO_{2}$$
(6)$$RO_{2}+NO\;\to \;RO+NO_{2}$$
(7)$$HO+RH+O_{2}\;\to \;RO_{2}+H_{2}O$$
(8)$$RO+O_{2}+hv\;\to \;HO_{2}+RCHO$$

If large amounts of NO_X_ are emitted, HO and RO_2_ react predominantly with NO_2_ (Equations (9) and (10)); if small amounts of NO_X_ are emitted, the free radical reaction dominates (Equations (11) and (12)). According to the formation mechanism of O_3_, the O_3_ concentration is closely related to the NO_X_ and VOCs concentrations because of the highly nonlinear relationship between O_3_ and its precursors. Therefore, it is more difficult to mitigate O_3_ than other pollutants.
(9)$$RO_{2}+NO_{2}\;\to \;RO_{2}NO_{2}$$
(10)$$HO+NO_{2}\;\to \;HNO_{3}$$
(11)$$HO_{2}+HO_{2}\;\to \;H_{2}O_{2}+O_{2}$$
(12)$$HO_{2}+RO_{2}\;\to \;RO_{2}H+O_{2}$$

The O_3_ concentration depends on the O_3_ formation process and diffusion [37,38,39]. Accordingly, the photochemical reaction rate [40], human activities, and meteorological conditions are the three dominant factors affecting the local O_3_ concentration [41,42]. Many studies have demonstrated that low cloudiness [43,44], intense solar radiation [45], high temperature [46,47], and low RH [48] can accelerate the O_3_ production rate [49,50]. High road network density [51], frequent motor vehicle braking, rapid acceleration, and high traffic flow [52] lead to high NO_X_ emissions [53]. Wind speed and direction can affect the horizontal distribution of O_3_ in local areas, and a low wind speed facilitates O_3_ accumulation [54,55].

Traffic emissions are the main pollutant source in urban areas. NO_X_ and VOC from traffic emissions are the main precursors of O_3_. Therefore, it is necessary to investigate the relationship between traffic conditions and O_3_ pollution. However, there are very few studies focusing on the influence of traffic situations on O_3_. We investigate the potential relationship between the O_3_ concentration and traffic conditions at roadside and urban background stations in Guangzhou, one of the largest cities in the PRD and China. The results provide a scientific reference for policymakers to establish emission mitigation policies.

## 2. Materials and Methods

### 2.1. Study Area and Measurement Data

Guangzhou is one of the largest cities in China, with a developed economy, dense population, and advanced manufacturing industries. The atmospheric pollutant concentrations were obtained from three national monitoring stations: two roadside stations (Yangji station (YJ station) and Huangsha station (HS station)) and one urban background station (Luhu station (LH station)) (Figure 1). The YJ station is located at an intersection of the main road (Zhongshan road) in the city center, about 5 m higher above ground. The HS station is located on a three-layer viaduct. The measurement instruments were installed between the second and third layers, about 20 m above the ground. The LH station is situated in Luhu Park, allowing us to compare air pollution in traffic and an urban park. The national measurement data were obtained from Guangzhou Ecological Environment Bureau (http://sthjj.gz.gov.cn/, accessed on 1 July 2021). The temporal resolution of the measurement data is one hour.

Meteorological data were obtained from Guangzhou Weather website (http://www.tqyb.com.cn/gz/weatherLive/autoStation/, accessed on 1 July 2021), including ambient temperature, wind speed, wind direction, solar radiation, and RH. The dynamic traffic data were obtained from the Guangzhou Municipal Bureau of Transportation (http://jtj.gz.gov.cn/jtcx/lkcx/, accessed on 1 July 2021). All the data were quality-controlled and covered the period from January to June 2021.

### 2.2. Analysis Approaches

A stepwise regression model was used to investigate the relationship between the potential impact factors and O_3_ concentration. Stepwise regression analysis automatically selects the most important variables to establish a predictive or explanatory model. The influencing factor are incorporated into the model one by one, and the statistical significance was evaluated. The insignificant factors were removed from the model.

## 3. Results and Discussion

### 3.1. Temporal Variations of NO_2_ and O_3_

#### 3.1.1. Daily Variations

Generally, pollutant concentrations are affected by several factors, such as emission sources, meteorological conditions, and pollutant formation mechanisms. The median diurnal variation of O_3_ and NO_2_ during the cold (from January to March) and warm (from April to June) seasons is shown in Figure 2. Similar diurnal patterns of O_3_ are observed at the three stations. The O_3_ concentration is low from 22:00 to the early morning on the following day. Then, it rapidly increases from around 8:00 in the morning and reaches the maximum around 14:00–16:00. As the solar radiation increases during the daytime, the O_3_ concentration increases [56,57] (Equations (1) and (2)). However, the O_3_ concentration remains low during the night. There are two reasons. First, less O_3_ is generated in the absence of sunlight. Second, NO can react with O_3_ to form NO_2_ and O_2_ during the night (Equation (4)), which is referred to as the titration effect of NO.

The diurnal variation of NO_2_ differs from that of O_3_. As shown in Figure 2d–f, the NO_2_ concentration is lower at 3:00–4:00 and 12:00–16:00 and higher at 6:00–8:00 and 20:00–22:00. The highest NO_2_ concentration occurs at 20:00–22:00. The NO_2_ concentration shows an increasing trend from 04:00–8:00 at the two roadside stations (HS and JY) because of traffic emissions. This increasing trend is not observed at the urban background station (LH). The solar radiation increases after 08:00. NO_2_ reacts with VOCs to produce O_3_, resulting in a decreasing trend at all three stations. The NO_2_ concentration increases after 16:00 due to lower solar radiation and a decrease in the photochemical reaction [58,59,60]. During the night, the NO_2_ concentration increases again due to the titration effect [61].

The seasonal difference in the pollutant concentration is larger for NO_2_ than for O_3_, as shown in Figure 2d–f. The NO_2_ concentration is higher in the cold season (from January to March) than in the warm season (from April to June). The decisive factor influencing the seasonal variation of the NO_2_ concentration is solar radiation. The average solar radiance in Guangzhou is 1352 kJ/ m^2^ in the cold season and 1806 kJ/ m^2^ in the warm season. Lower solar radiation leads to less O_3_ generation and less NO_2_ consumption. Another possible factor may be the lower RH in winter. In Guangzhou, the average RH is 59.04% and 86.2% in the cold and warm seasons, respectively [62,63]. A higher RH results in a stronger photochemical reaction and a lower NO_2_ concentration in the warm season. Another possible explanation is the seasonal change in the planetary boundary layer height. It is 717 m in winter and 1239 m in summer in Guangzhou [64,65]. A lower planetary boundary layer accumulates NO_2_, resulting in a higher NO_2_ concentration [66]. However, the seasonal difference in the O_3_ concentration is smaller than that of the NO_2_ concentration. The reason is that O_3_ is a secondary pollutant whose concentration is controlled by highly complex and nonlinear secondary formation mechanisms.

#### 3.1.2. Weekly Variations

The weekly variations in the O_3_ and NO_2_ concentrations at the three stations are illustrated in Figure 3. In general, the weekly trends of the O_3_ and NO_2_ concentrations are similar at three stations, but the average concentrations are different. As shown in Figure 3a, the O_3_ concentration is significantly higher on weekends (Saturday and Sunday) than on weekdays (from Monday to Friday), indicating the weekend effect of O_3_. It is believed to be related to a change in the proportion of O_3_ precursor emissions and other pollutant emissions from human activities [67]. Fewer human activities on weekends lead to lower PM_2.5_ and a lower aerosol optical thickness and radiation extinction. Therefore, the O_3_ concentrations are higher on the weekend than on weekdays due to stronger photochemical reactions [68,69]. Moreover, high traffic flow during the morning rush hour results in a rapid increase in the NO concentration, inhibiting O_3_ formation on weekdays [70,71].

Differences in the O_3_ concentration are observed at the three stations. The highest O_3_ concentration was measured at the LH station, followed by the two roadside stations YJ and HS. The reason is the surrounding environment. The LH station is located in Luhu Park. VOCs generated by biological sources compete with NO, reducing the inhibition of NO on O_3_ and leading to a higher O_3_ concentration [72,73]. The YJ station is surrounded mostly by business and entertainment areas with frequent human activities. Large amounts of NO_X_ are emitted from traffic inhibited O_3_ formation. In addition, the titration effect of NO is stronger at the YJ station, leading to a slightly lower O_3_ concentration at the YJ station than at the LH station. The HS station is a roadside station located near a park. It has higher vegetation cover than the YJ station.

The weekly variation in the NO_2_ concentration shows a significantly different pattern than that of the O_3_ concentration. The NO_2_ concentration is slightly higher on weekdays than on the weekend due to higher anthropogenic emissions, especially traffic emissions in urban areas [74,75,76,77]. The NO_2_ concentration is the highest at the HS station, followed by the YJ and LH stations, which is consistent with the traffic emissions and the local environment of the three stations.

### 3.2. Influencing Factors

#### 3.2.1. Synergistic Variation of O_3_ and NO_2_

Figure 4 shows the scatterplots of the O_3_ and NO_2_ concentrations during the daytime (07:00–19:00) and nighttime (20:00–06:00) at the three stations. The linear regression model has a negative slope for all three stations during the daytime and nighttime, indicating that the NO_2_ concentration decreases as the O_3_ concentration increases. However, differences are observed between daytime and nighttime. In the daytime, NO_2_ is consumed, and O_3_ is produced (Equations (2) and (3)). However, without a photochemical reaction during nighttime, O_3_ is converted to NO_2_ due to the titration effect (Equation (4)), leading to a lower O_3_ concentration. Due to the highly nonlinear and complex O_3_ formation mechanism, the R^2^ value is low for all fitting results. The R^2^ value is larger during nighttime at all three stations due to the absence of the photochemical reaction, the titration effect of NO, and weaker vertical diffusion [78,79]. The nighttime fitting degree is better at the LH station than at the roadside stations. The reason might be the surrounding environment of the LH station. The vegetation cover is higher; thus, vegetation respiration is stronger at night. Consequently, the NO_2_ and O_3_ concentrations are relatively stable, leading to a better fitting degree.

The fitted results of the three stations are similar. However, the dominant emission sources differ at the three stations. This result indicates no significant effect of traffic emissions on the O_3_ concentration at the roadside stations. Due to the absence of VOCs, a dynamic equilibrium exists between O_3_ and NO_X_ in the atmosphere. Thus, O_3_ is not accumulated and does not exceed the air pollution standard [80,81]. However, the reaction between VOCs and NO weakens the inhibitory effect of NO on O_3_, resulting in high O_3_ pollution [82]. Controlling NO_X_ emissions does not mitigate O_3_ pollution. Moreover, Guangzhou is in the VOC-limitation area [83,84]. Limiting vehicle emissions to reduce the NO_X_ concentration may even increase the O_3_ concentration. Therefore, the focus should be on industrial, energy, and transportation emission mitigation and the influence of meteorological conditions to minimize O_3_ pollution.

#### 3.2.2. Pearson Correlation and Stepwise Regression Analyses

Pearson correlation analysis and stepwise regression analysis were conducted to describe the relationship between the pollutant concentration and other factors, such as meteorological parameters and dynamic traffic parameters. Table 1 and Table 2 show the results of Pearson’s correlation analysis and stepwise regression analysis, respectively. Pearson’s correlation shows the correlation between the O_3_ concentration and potential factors, and the stepwise regression model determines the significant impact factors. The beta values are used to quantify the contribution of the variables. Briefly, the O_3_ concentration is positively correlated with solar radiation, temperature, and travel-time ratio and negatively correlated with the NO_2_ concentration, wind speed, and vehicle speed (Table 1). The stepwise regression model shows that the significant factors affecting O_3_ concentration are temperature, NO_2_ concentration, and RH. As shown in Table 1, the O_3_ concentration positively correlates with the travel-time ratio. The travel time ratio is the ratio of the actual travel time to the ideal travel time in smooth traffic flow. The larger the ratio, the higher the degree of traffic congestion. The NO_X_ and VOC emissions are higher during frequent vehicle braking than during uniform driving. Thus, more O_3_ precursors are emitted, leading to a significant positive correlation between O_3_ concentration and travel-time ratio. The temperature is positively correlated with O_3_ concentration as a result of O_3_ formation. The negative correlation between the NO_2_ and O_3_ concentrations has already been discussed in Section 3.2.1. Moreover, a negative correlation is observed between vehicle speed and O_3_ concentration. The fuel consumption is higher at higher speeds than at lower speeds, resulting in more precursor emissions and a higher O_3_ concentration. Wind speed and O_3_ concentration are negatively correlated because of the dilution effect. The O_3_ concentration is lower at higher RH due to wet deposition. Moreover, an increase in RH significantly reduces the number of oxygen atoms, reducing the amount of O_3_ generation.

The secondary pollutant O_3_ is correlated with several factors. The vehicle speed and travel-time ratio are significantly correlated with the O_3_ concentration, indicating the importance of traffic emissions on O_3_ pollution in urban areas.

#### 3.2.3. Case Study

As discussed in the previous section, traffic emissions affect the O_3_ concentration but are not the dominant factor. Many studies demonstrated that solar radiation was a significant factor influencing O_3_ formation. A case study was conducted to quantify the influence of solar radiation on O_3_ concentration in Guangzhou. Two weeks were selected: 1 February to 7 February 2021, with sunny weather, and 24 February to 2 March 2021, with cloudy weather.

The pollutant concentrations and related parameters are listed in Table 3. The O_3_ concentration is substantially different on sunny and cloudy days at all three stations, indicating the predominant influence of solar radiation. The O_3_ concentration is 2–3 times higher on sunny days than on cloudy days in the daytime and nighttime. However, there are no large differences in the NO_2_ concentration. In the daytime, there are no differences in the NO_2_ concentration between sunny and cloudy days. However, the nighttime NO_2_ concentration is 1.5 to 2 times higher on sunny days than on cloudy days. More O_3_ is formed during sunny days, leading to a stronger titration effect and a higher NO_2_ concentration during the nighttime on sunny days. It should be noted that the NO_2_ concentration is lower at the LH station than at the two roadside stations during the daytime. However, the O_3_ concentration is similar at all three stations due to the lower inhibitory effect of NO, as discussed in Section 3.1.2. This finding confirms our results, i.e., traffic emissions contribute significantly to O_3_ generation, but the contribution is not higher at roadside stations than at the urban background station.

The scatterplots of the NO_2_ and O_3_ concentrations in the two periods at YJ and HS are shown in Figure 5. The colored dots indicate the dynamic traffic conditions. The linear regression results demonstrate that the negative correlation between the NO_2_ and O_3_ concentrations is stronger during the daytime than during the nighttime at both stations due to the stronger photochemical reaction strength. Furthermore, no significant relationship is observed between the O_3_ concentration and dynamic traffic conditions.

## 4. Conclusions

This study evaluated the factors influencing the O_3_ concentration in traffic and urban background environments. The diurnal and weekly variation of the O_3_ and NO_2_ concentrations demonstrated a similar pattern at the three stations. These results were attributed to differences in the O_3_ generation mechanism, meteorological conditions, and emission sources. However, no significant differences in the O_3_ variation were observed between the three stations, implying that the O_3_ concentration was not significantly higher in the traffic environment than in the urban background environment. Since Guangzhou is located in a VOC-limited area, the lower O_3_ concentration in the urban background area is due to the lower inhibition of NO on O_3_.

Pearson correlation analysis and stepwise regression analysis were used to describe the relationship between the pollutant concentration and the influencing factors, such as meteorological and dynamic traffic parameters. Traffic and meteorological parameters (temperature, solar radiation, RH, and precipitation) were significantly correlated with the O_3_ concentration at the two roadside stations. It was concluded that traffic emissions contributed to O_3_ pollution in the urban area but were not the decisive factor, while the meteorological factors also influenced the O_3_ concentration.

A case study was conducted for two weeks to quantify the influence of solar radiation on O_3_ concentration in Guangzhou. On sunny days, the O_3_ concentration exceeded 90 μg/m^3^ at the three sites. It was 2 to 3 times higher than during cloudy days due to meteorological conditions. The dynamic traffic condition (travel-time ratio) had no significant relationship with the O_3_ and NO_2_ concentrations at the two roadside stations.

This study analyzed the temporal variation of O_3_ and its precursor NO_2_ at roadside and urban background environments in Guangzhou and its influencing factors. The results confirmed that limiting traffic emissions might increase O_3_ concentrations in Guangzhou. Therefore, emission mitigation should be performed, i.e., industrial, energy, and transportation emission mitigation, and the influence of meteorological conditions should be considered to minimize O_3_ pollution. However, some limitations exist in this study. Due to a lack of NO and VOCs data, the relationship between O_3_ concentration and NO and VOCs was not analyzed. In future, a mobile measurement focusing on O_3_ will be carried out in Guangzhou, and a more detailed analysis will be performed.

## Figures and Tables

**Figure 1 ijerph-19-12961-f001:**
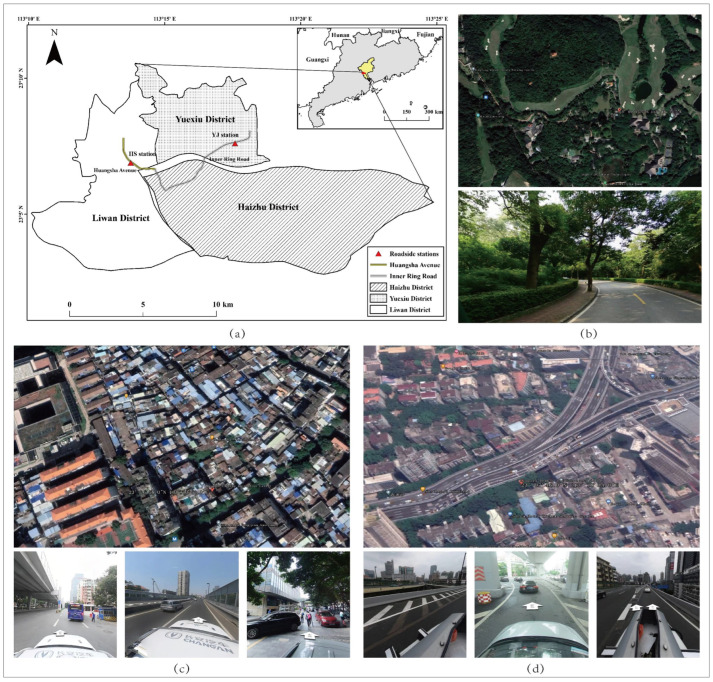
Overview of the study area and atmospheric monitoring stations. (**a**) Location of three stations; (**b**) Luhu station (LH); (**c**) Huangsha station (HS); (**d**) Yangji station (YJ).

**Figure 2 ijerph-19-12961-f002:**
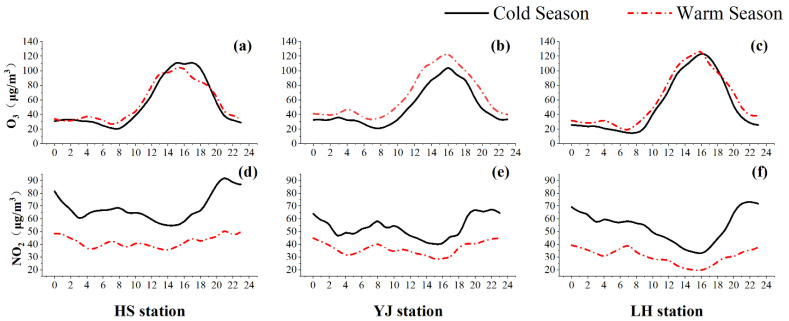
Diurnal variation of typical pollutants in cold and warm seasons: O_3_ concentrations at (**a**) HS station, (**b**) YJ station, and (**c**) LH station; NO_2_ concentrations at (**d**) HS station, (**e**) YJ station, and (**f**) LH station.

**Figure 3 ijerph-19-12961-f003:**
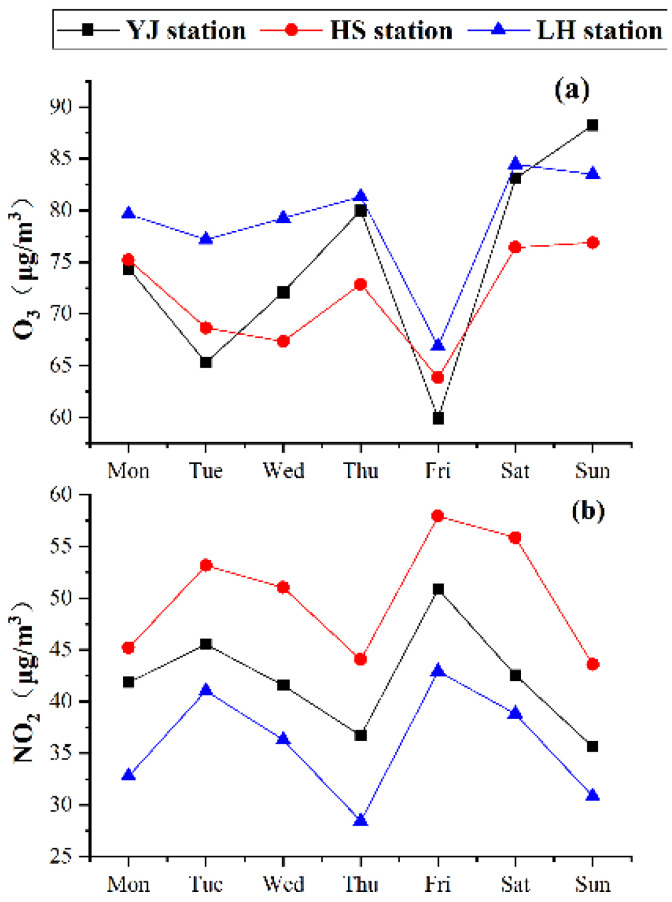
Weekly variation in O_3_ (**a**) and NO_2_ (**b**) concentrations at the three stations.

**Figure 4 ijerph-19-12961-f004:**
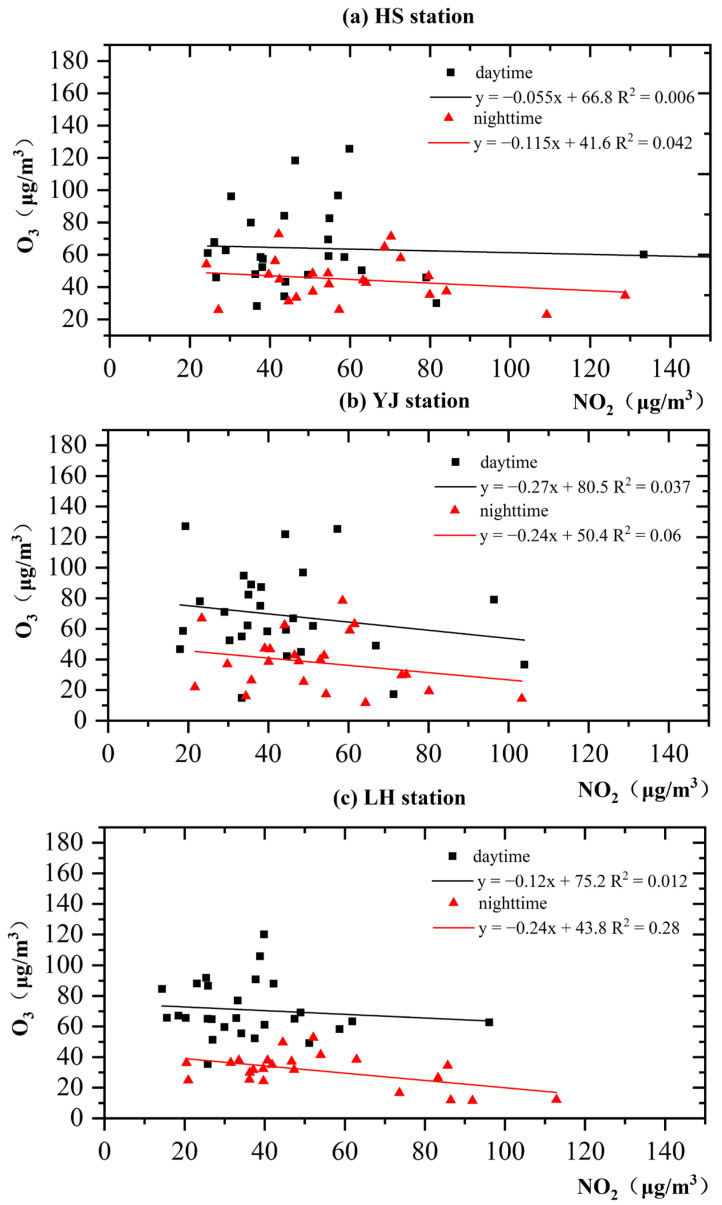
Scatterplot of O_3_ and NO_2_ concentrations at HS (**a**), YJ (**b**), and LH (**c**).

**Figure 5 ijerph-19-12961-f005:**
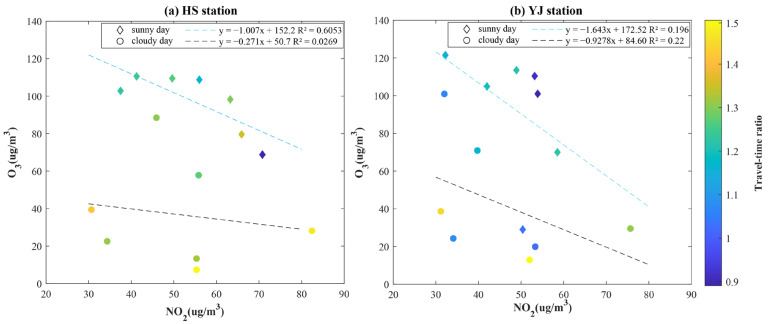
Scatterplot of daily O_3_ and NO_2_ concentrations in the two periods at the HS (**a**) and YJ (**b**) stations. The colored dots denote the travel-time ratio.

**Table 1 ijerph-19-12961-t001:** Pearson’s correlation coefficients between O_3_ concentration and various factors.

Impact Factors	Daytime	Nighttime
Temperature (°C)	0.047 ******	0.057 ******
Wind speed (m/s)	−0.082 **	−0.057 ******
Daily precipitation (mm)	**−0.101 ****	−0.006
Vehicle speed (m/s)	**−0.111 ****	**−0.111 ****
Travel-time ratio	**0.150 ****	**0.129 ****
NO_2_ (μg/m^3^)	−0.220 *	−0.153 **
RH (%)	**−0.495 ****	**−0.226 ****
Solar radiation (J/m^2^)	**0.448 ****	**0.279 ****

** Significant at the 0.01 level. * Significant at the 0.05 level.

**Table 2 ijerph-19-12961-t002:** Results of stepwise regression model between O_3_ concentration and various factors.

Model	Daytime	*p*	Nighttime	*p*
	Beta Value		Beta Value	
Temperature (°C)	0.386	0.000	0.207	0.000
Wind speed (m/s)	−0.076	0.000	−0.124	0.000
Daily precipitation (mm)	0.092	0.000	0.036	0.037
Vehicle speed (m/s)	−0.077	0.000	−0.063	0.000
NO_2_ (μg/m^3^)	−0.407	0.000	−0.611	0.000
RH (%)	−0.578	0.000	−0.389	0.000
Solar radiation (J/m^2^)	-	-	0.182	0.000

The dependent variable: O_3_ (μg/m^3^).

**Table 3 ijerph-19-12961-t003:** The pollutant concentrations and related parameters in the two periods.

Period	Station	O_3_ (μg/m^3^)		NO_2_ (μg/m^3^)		Travel-Time Ratio		Solar Radiation (KJ/m^2^)	RH (%)
		Day Time	Night Time	Day Time	Night Time	Day Time	Night Time		
Sunny days	HS	97.43	52.95	54.28	79.05	1.14	1.03	17,627.04	64.71%
	JY	94.70	63.45	48.80	63.66	1.25	1.06		
	LH	102.31	45.88	38.87	79.34	-	-		
Cloudy days	HS	37.30	21.15	52.34	53.10	1.21	1.05	10,300.89	75.08%
	JY	42.90	27.86	46.16	48.26	1.29	1.08		
	LH	41.70	25.75	36.99	41.59	-	-		

## Data Availability

The data presented in this study are available on request from the corresponding website.

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
