# Peer review of "Factors Influencing O3 Concentration in Traffic and Urban Environments: A Case Study of Guangzhou City"

_ijerph, 2022, doi:10.3390/ijerph191912961_

Round 1
Reviewer 1 Report
THis is an interesting review and investigation of the pollution levels in a busy city, the interpretation of the findings appear correct but it would be good to understand in more detail what this adds to evidence that is already available?
The statistics appear approapriate
The whole paper would benefit from an english language review
Author Response
Response:
Thanks for your suggestions. We have revised in detail and added some content.
Many thanks for the corrections and suggestions. To quantify the influence of solar radiation on O3 concentration in Guangzhou, a new section “3.2.3. Case study” has been added in the manuscript. At the same time, we added solar radiation and relative humidity (RH) data, and reanalysis with Pearson correlation model and stepwise regression model (Table 1. and Table 2. in the manuscript). We have added a comparative analysis of O3, NO2, travel-time ratio, solar radiation and RH with different weather conditions (Table 3. in the manuscript). We found that the O3 concentration is 2-3 times higher on sunny days than on cloudy days in the daytime and nighttime. However, the nighttime NO2 concentration is 1.5 to 2 times higher on sunny days than on cloudy days.
Table 1. Pearson’s correlation coefficients between O3 concentration and various factors.
Impact Factors |
Daytime |
Nighttime |
Temperature (℃) |
.047** |
.057** |
Wind speed (m/s) |
-.082** |
-.057** |
Daily precipitation (mm) |
-.101** |
-.006 |
Vehicle speed (m/s) |
-.111** |
-.111** |
Travel-time ratio |
.150** |
.129** |
NO2 (ug/m3) |
-.220* |
-.153** |
RH (ï¼…) |
-.495** |
-.226** |
Solar radiation (J/m2) |
.448** |
.279** |
** Significant at the 0.01 level. * Significant at the 0.05 level.
Table 2. Results of stepwise regression model between O3 concentration and various factors.
Impact Factors |
Daytime |
P |
Night-time |
P |
Beta value |
Beta value |
|||
Temperature (℃) |
.386 |
.000 |
.207 |
.000 |
Wind speed (m/s) |
-.076 |
.000 |
-.124 |
.000 |
Daily precipitation (mm) |
.092 |
.000 |
.036 |
.037 |
Vehicle speed (m/s) |
-.077 |
.000 |
-.063 |
.000 |
NO2 (ug/m3) |
-.407 |
.000 |
-.611 |
.000 |
RH (ï¼…) |
-.578 |
.000 |
-.389 |
.000 |
Solar radiation (J/m2) |
--- |
--- |
.182 |
.000 |
- The dependent variable: O3 (μg/m3).
Table 3. The pollutant concentrations and related parameters in the two periods.
Case period |
Station |
O3 (ug/m3) |
NO2 (ug/m3) |
Travel-time ratio |
Solar radiation (J/ m2) |
RH (%) |
||||
Day time |
Night time |
Day time |
Night time |
Day time |
Night time |
|||||
Sunny days |
HS |
97.43 |
52.95 |
54.28 |
79.05 |
1.14 |
1.03 |
17627040.45 |
64.71% |
|
JY |
94.70 |
63.45 |
48.80 |
63.66 |
1.25 |
1.06 |
||||
LH |
102.31 |
45.88 |
38.87 |
79.34 |
--- |
--- |
||||
Cloudy days |
HS |
37.30 |
21.15 |
52.34 |
53.10 |
1.21 |
1.05 |
10300885.32 |
75.08% |
|
JY |
42.90 |
27.86 |
46.16 |
48.26 |
1.29 |
1.08 |
||||
LH |
41.70 |
25.75 |
36.99 |
41.59 |
--- |
--- |
||||
The scatterplots of the NO2 and O3 concentrations in the two periods at YJ and HS are shown in Fig. 5. The colored dots indicate the dynamic traffic conditions. The linear regression results show that the negative correlation between the NO2 and O3 concentrations is stronger during the daytime than during the nighttime at both stations due to the stronger photochemical reaction strength. Furthermore, no significant relationship is observed between the O3 concentration and dynamic traffic conditions.
Figure 5. Scatterplot of daily O3 and NO2 concentrations in the two periods at the HS (a) and YJ (b) stations. The colored dots denote the travel-time ratio.
Most parts of the manuscript have been rewritten. The manuscript has been also edited by Editor Bar Language Editing to improve the language.
Please see the attachment.

Reviewer 2 Report
O3 variation characteristics and influencing factors of typical traffic arteries based on NO2 Cooperation: A case study of two roadside stations in Guangzhou City. This topic is suitable for publication in this journal. However, there are some important information is missing and several issues need to be revised. Therefore, I would like to recommend publication with 'major revision'.
1.line 17, Traffic emission is an important source of O3 emission,traffic is source of NOx, VOCs, not O3.
2. line 18-19, many national control stations locate at downtown, and near the traffic emission.
3. abstract need to be re-written. Please add more quantitative results. The abstract should explicitly include the key finding of your manuscript.
4. line 42, O3 mainly from photochemical, not the stationary and mobile sources. What are the stationary sources?
5. line 99-102, add the references.
6. introduction, It is necessary to supplement some researches in recent years:
https://doi.org/10.1088/1748-9326/ac69fe
https://doi.org/10.1016/j.scitotenv.2021.152449
https://doi.org/10.1016/j.envres.2022.114095
https://doi.org/10.1016/j.scitotenv.2021.148474
https://doi.org/10.1016/j.apr.2021.101247
7. There are too many grammar problems in this article, please modify them in detail.
8. Results and Discussion, It is necessary to carry out a detailed analysis in combination with your own data, and give more quantitative results. In addition, the explanation for the cause of the phenomenon must be based on the basis and specific data, rather than arbitrary explanations.
9. line 187-189,different monitoring stations. What's the difference? Specific description is required.
10.line 218-219,Add data on the height of the boundary layer, how much is the lower?
11.line 219-220, the short sunshine hours, low temperature, stable and long duration of the inversion layer in winter. How many hours are short in winter? And how much low?
12.line 221-226, need to add the data of humidity and solar radiation.
13. line 231-236, need to add the specific values of data.
14. line 244-255, need to add the specific values of data.
15. 3.1. Variation characteristics of typical pollutants at different time scales, Add comparative analysis of three stations.
16. line 329-331, There is no VOCs data in this paper.
17. line 350-365, The discussion about VOCs needs to be deleted.
18. 3.2. Study of influence factors. Add comparative analysis of three stations.
19. Conclusions and Foresight, need to be re-written, needs to focus on giving quantitative results. It’s too long now.
Author Response
Response to comments of referee #2
(1) Line 17, Traffic emission is an important source of O3 emission,traffic is source of NOx, VOCs, not O3.
Response:
Thanks for your suggestions. We have rewritten the abstract and this sentence has been removed.
In the revised manuscript, the sentence has been revised as “NOX and volatile organic compounds (VOCs) from traffic emissions are the main precursors of O3”
(2) Line 18-19, many national control stations locate at downtown, and near the traffic emission.
Response:
Thanks for the suggestion. To avoid ambiguity, this sentence has been removed.
The national control stations mainly monitor the overall condition of the urban air environment, so the station is 8 to 15 m above the ground. The roadside stations are used to monitor the impact of motor vehicle exhaust on air quality on the road, more concerned about the primary source emissions, the height from the ground is only 2 m.
(3) Abstract need to be re-written. Please add more quantitative results. The abstract should explicitly include the key finding of your manuscript.
Response:
Thanks for the comment. We have rewritten this part.
In the revised manuscript, we add more quantitative results and clarify the key content of the article. Here are the changes:
“Ozone (O3) pollution is a serious issue in China, posing a significant threat to people's health. Traffic emissions are the main pollutant source in urban areas. NOX and volatile organic compounds (VOCs) from traffic emissions are the main precursors of O3. Thus, it is crucial to investigate the relationship between traffic conditions and O3 pollution. This study focused on the potential relationship between O3 concentration and traffic conditions at roadside and urban background in Guangzhou, one of the largest cities in China. The results showed that no significant difference in the O3 concentration was observed between roadside and urban background environments. However, the O3 concentration was 2 to 3 times higher on sunny days (above 90 μg/m3) than on cloudy days due to meteorological conditions. The results confirmed that limiting traffic emissions may increase O3 concentrations in Guangzhou. Therefore, the focus should be on industrial, energy, and transportation emission mitigation and the influence of meteorological conditions to minimize O3 pollution. The results in this study provide some theoretical basis for mitigation emission policies in China.”
(4) Line 42, O3 mainly from photochemical, not the stationary and mobile sources. What are the stationary sources?
Response:
Thanks for your comment. We have rewritten this sentence and added some relevant references.
In the revised manuscript, the sentence has been revised as “There are three major sources of near-ground O3 precursors: traffic emissions, industry emissions and emissions by power plants.”
(5) Line 99-102, add the references.
Response:
Many thanks for the corrections and suggestions. Some of the references have been added in the manuscript.
(6) Introduction, It is necessary to supplement some researches in recent years:
https://doi.org/10.1088/1748-9326/ac69fe
https://doi.org/10.1016/j.scitotenv.2021.152449
https://doi.org/10.1016/j.envres.2022.114095
https://doi.org/10.1016/j.scitotenv.2021.148474
https://doi.org/10.1016/j.apr.2021.101247
Response:
Many thanks for the corrections and suggestions. We have added some researches in recent years.
(7) There are too many grammar problems in this article, please modify them in detail.
Response:
Many thanks for the corrections and suggestions. Most parts of the manuscript have been rewritten. The manuscript has been also edited by Editor Bar Language Editing to improve the language.
(8) Results and Discussion, It is necessary to carry out a detailed analysis in combination with your own data, and give more quantitative results. In addition, the explanation for the cause of the phenomenon must be based on the basis and specific data, rather than arbitrary explanations.
Response:
Many thanks for the corrections and suggestions. To quantify the influence of solar radiation on O3 concentration in Guangzhou, a new section “3.2.3. Case study” has been added in the manuscript. At the same time, we added solar radiation and relative humidity (RH) data, and reanalysis with Pearson correlation model and stepwise regression model (Table 1. and Table 2. in the manuscript). We have added a comparative analysis of O3, NO2, travel-time ratio, solar radiation and RH with different weather conditions (Table 3. in the manuscript). We found that the O3 concentration is 2-3 times higher on sunny days than on cloudy days in the daytime and nighttime. However, the nighttime NO2 concentration is 1.5 to 2 times higher on sunny days than on cloudy days.
Table 1. Pearson’s correlation coefficients between O3 concentration and various factors.
Impact Factors |
Daytime |
Nighttime |
Temperature (℃) |
.047** |
.057** |
Wind speed (m/s) |
-.082** |
-.057** |
Daily precipitation (mm) |
-.101** |
-.006 |
Vehicle speed (m/s) |
-.111** |
-.111** |
Travel-time ratio |
.150** |
.129** |
NO2 (ug/m3) |
-.220* |
-.153** |
RH (ï¼…) |
-.495** |
-.226** |
Solar radiation (J/m2) |
.448** |
.279** |
** Significant at the 0.01 level. * Significant at the 0.05 level.
Table 2. Results of stepwise regression model between O3 concentration and various factors.
Impact Factors |
Daytime |
P |
Night-time |
P |
Beta value |
Beta value |
|||
Temperature (℃) |
.386 |
.000 |
.207 |
.000 |
Wind speed (m/s) |
-.076 |
.000 |
-.124 |
.000 |
Daily precipitation (mm) |
.092 |
.000 |
.036 |
.037 |
Vehicle speed (m/s) |
-.077 |
.000 |
-.063 |
.000 |
NO2 (ug/m3) |
-.407 |
.000 |
-.611 |
.000 |
RH (ï¼…) |
-.578 |
.000 |
-.389 |
.000 |
Solar radiation (J/m2) |
--- |
--- |
.182 |
.000 |
- The dependent variable: O3 (μg/m3).
Table 3. The pollutant concentrations and related parameters in the two periods.
Case period |
Station |
O3 (ug/m3) |
NO2 (ug/m3) |
Travel-time ratio |
Solar radiation (J/ m2) |
RH (%) |
||||
Day time |
Night time |
Day time |
Night time |
Day time |
Night time |
|||||
Sunny days |
HS |
97.43 |
52.95 |
54.28 |
79.05 |
1.14 |
1.03 |
17627040.45 |
64.71% |
|
JY |
94.70 |
63.45 |
48.80 |
63.66 |
1.25 |
1.06 |
||||
LH |
102.31 |
45.88 |
38.87 |
79.34 |
--- |
--- |
||||
Cloudy days |
HS |
37.30 |
21.15 |
52.34 |
53.10 |
1.21 |
1.05 |
10300885.32 |
75.08% |
|
JY |
42.90 |
27.86 |
46.16 |
48.26 |
1.29 |
1.08 |
||||
LH |
41.70 |
25.75 |
36.99 |
41.59 |
--- |
--- |
||||
The scatterplots of the NO2 and O3 concentrations in the two periods at YJ and HS are shown in Fig. 5. The colored dots indicate the dynamic traffic conditions. The linear regression results show that the negative correlation between the NO2 and O3 concentrations is stronger during the daytime than during the nighttime at both stations due to the stronger photochemical reaction strength. Furthermore, no significant relationship is observed between the O3 concentration and dynamic traffic conditions.
Figure 5. Scatterplot of daily O3 and NO2 concentrations in the two periods at the HS (a) and YJ (b) stations. The colored dots denote the travel-time ratio.
(9) Line 187-189,different monitoring stations. What's the difference? Specific description is required.
Response:
Many thanks for the corrections and suggestions. This sentence has been removed to avoid ambiguity. In this section, we focus on comparing the daily variation of pollutant concentrations at different types of stations, and in the following we will clarify "the roadside and urban background stations".
(10) line 218-219, Add data on the height of the boundary layer, how much is the lower?
Response:
Thanks for the comment. We have specified the planetary boundary layer height for the cold and warm seasons in Guangzhou and added related references.
“Another possible explanation is the seasonal change in the planetary boundary layer height. It is 717 m in winter and 1239 m in summer in Guangzhou”
(11) line 219-220, the short sunshine hours, low temperature, stable and long duration of the inversion layer in winter. How many hours are short in winter? And how much low?
Response:
Thanks for the comment. We have combined specific parameters and relevant references for in-depth analysis. Meanwhile we have rewritten this section and the corresponding sentence has been removed.
(12) line 221-226, need to add the data of humidity and solar radiation.
Response:
Thanks for the comment. In order to increase the accuracy, we have added specific values.
“The decisive factor influencing the seasonal variation of the NO2 concentration is solar radiation. The average solar radiance in Guangzhou is 1352 kJ/ m2 in the cold season and 1806 kJ/ m2 in the warm season. In Guangzhou, the average RH is 59.04% and 86.2% in the cold and warm seasons, respectively.”
(13) line 231-236, need to add the specific values of data.
Response:
Thank you for your reminder, this section has been removed to clarify the key to the article and reduce redundancy.
(14) line 244-255, need to add the specific values of data.
Response:
Thank you for your reminder, we have added a specific data analysis section to clarify the key points of the article and to reduce redundancy the sentences has been removed.
(15) 3.1. Variation characteristics of typical pollutants at different time scales, Add comparative analysis of three stations.
Response:
Thanks for the comment. We have added some comparative descriptions of the three sites in the article.
In the Fig. 2(a)-(c), it can be found that the daily variation of O3 concentration at the three stations is similar compared to other pollutants. Meanwhile, combining with Fig. 2(d)-(f), we added a comparative analysis of NO2 concentrations at the three stations.
(16) line 329-331, There is no VOCs data in this paper.
Response:
Thanks for the comment. Because studies by other scholars have verified this conclusion, we added relevant literature to increase the credibility.
(17) line 350-365, The discussion about VOCs needs to be deleted.
Response:
Thanks for the comment. We have rewritten this part and added related studies.
“This result indicates no significant effect of traffic emissions on O3 concentrations at the roadside stations. Due to the absence of VOCs, a dynamic equilibrium exists between O3 and NOX in the atmosphere. Thus, O3 is not accumulated and does not exceed the air pollution standard. However, the reaction between VOCs and NO weakens the inhibitory effect of NO on O3, resulting in high O3 pollution. Controlling NOX emissions does not mitigate O3 pollution. Moreover, Guangzhou is in the VOC-limitation area. Limiting vehicle emissions to reduce the NOX concentration may even increase the O3 concentration. Therefore, the focus should be on industrial, energy, and transportation emission mitigation and the influence of meteorological conditions to minimize O3 pollution.”
(18) 3.2. Study of influence factors. Add comparative analysis of three stations.
Response:
Thanks for the comment. We have rewritten this part and added comparative analysis of three sites. We added solar radiation and RH data and reanalysis with Pearson correlation model and stepwise regression model (Table 1. and Table 2. in the manuscript). At the same time, we have added case studies (3.2.3. Case study in the manuscript) of three sites (Table 3. and Fig. 5 in the manuscript).
(19) Conclusions and Foresight, need to be re-written, needs to focus on giving quantitative results. It’s too long now.
Response:
Thank you for your reminder. We have rewritten the entire paragraph. We have streamlined the conclusion and highlighted the quantitative results.
Please see the attachment.

Reviewer 3 Report
This study investigates O3, NO2 and PM2.5 variation patterns and their influencing factors at roadside and background stations in Guangzhou and explores whether O3 concentration can be effectively reduced by limiting traffic emissions. The following comments should be considered before this manuscript can be accepted for publication.
1. The main purpose of this study is to investigate O3 variation characteristics and influencing factors of typical traffic arteries based on NO2 Cooperation. I suggest focusing on O3 characteristic and formation mechanism and remove the content of PM2.5.
2. Line 16. “Ozone (O3) is the primary pollutant in the Pearl River Delta, posing a great threat to people's health.” O3 is “secondary” air pollutant. I suggest changing “primary” to “main” or “important” in order to avoid misunderstanding.
3. Line 17. “Traffic emission is an important source of O3 emission and is studied by scholars globally.” Similar to the previous comment, traffic emission does not produce O3 directly. Vehicles emit precursor air pollutants which result in formation of O3. Please rewrite this sentence.
4. One of the main purpose of this study is to explore whether O3 concentration can be effectively reduced by limiting traffic emissions. Please add the related results in Abstract.
5. Only O3 data was obtained for this research. How can the formation mechanism be explored clearly without NOx data?
6. The authors emphasize that this study used data from roadside but not data from national monitoring station which is far from traffic arteries. The reaction rate of photochemical formation of O3 should be discussed in this study.
Author Response
Response to comments of referee #3
(1) The main purpose of this study is to investigate O3 variation characteristics and influencing factors of typical traffic arteries based on NO2 Cooperation. I suggest focusing on O3 characteristic and formation mechanism and remove the content of PM2.5.
Response:
Thanks for your suggestions. We have rewritten the entire paragraph and removed the part of PM2.5.
(2) Line 16. “Ozone (O3) is the primary pollutant in the Pearl River Delta, posing a great threat to people's health.” O3 is “secondary” air pollutant. I suggest changing “primary” to “main” or “important” in order to avoid misunderstanding.
Response:
Thanks for your suggestions. We have rewritten the abstract section. In the revised manuscript, the sentence has been revised as “Ozone (O3) pollution is a serious issue in China, posing a significant threat to people's health.”
(3) Line 17. “Traffic emission is an important source of O3 emission and is studied by scholars globally.” Similar to the previous comment, traffic emission does not produce O3 directly. Vehicles emit precursor air pollutants which result in formation of O3. Please rewrite this sentence.
Response:
Thanks for your suggestions. We have rewritten the abstract and this sentence has been removed.
In the revised manuscript, the sentence has been revised as “NOX and volatile organic compounds (VOCs) from traffic emissions are the main precursors of O3.”
(4) One of the main purpose of this study is to explore whether O3 concentration can be effectively reduced by limiting traffic emissions. Please add the related results in Abstract.
Response:
Thanks for your suggestions. We have rewritten the abstract.
“Ozone (O3) pollution is a serious issue in China, posing a significant threat to people's health. Traffic emissions are the main pollutant source in urban areas. NOX and volatile organic compounds (VOCs) from traffic emissions are the main precursors of O3. Thus, it is crucial to investigate the relationship between traffic conditions and O3 pollution. This study focused on the potential relationship between O3 concentration and traffic conditions at roadside and urban background in Guangzhou, one of the largest cities in China. The results showed that no significant difference in the O3 concentration was observed between roadside and urban background environments. However, the O3 concentration was 2 to 3 times higher on sunny days (above 90 μg/m3) than on cloudy days due to meteorological conditions. The results confirmed that limiting traffic emissions may increase O3 concentrations in Guangzhou. Therefore, the focus should be on industrial, energy, and transportation emission mitigation and the influence of meteorological conditions to minimize O3 pollution. The results in this study provide some theoretical basis for mitigation emission policies in China.”
(5) Only O3 data was obtained for this research. How can the formation mechanism be explored clearly without NOx data?
Response:
Thanks for your suggestions. This study has some limitations. Due to a lack of NO and VOCs data, the relationship between the concentrations of O3 and its precursors was not analyzed. Moreover, measurements of the O3 concentration in Guangzhou with a higher spatial resolution will be carried out, and a more detailed analysis will be performed in future studies. This section has been added at the end of the conclusion as a limitation
(6) The authors emphasize that this study used data from roadside but not data from national monitoring station which is far from traffic arteries. The reaction rate of photochemical formation of O3 should be discussed in this study.
Response:
Thank you for your reminder. Due to the complexity of photochemical reactions and the variety of reactants, it is difficult to quantify the rate of photochemical reactions, so we have added relevant studies by other scholars in the manuscript.
Please see the attachment.

Round 2
Reviewer 1 Report
This paper has had an extensive re-write and as such is much is easier to read. The results are now displayed in a more accessible manner
Reviewer 2 Report
accept.
Reviewer 3 Report
This revised version can be accepted.